# The Effect of Augmented Reality-Based Proprioceptive Training Program on Balance, Positioning Sensation and Flexibility in Healthy Young Adults: A Randomized Controlled Trial

**DOI:** 10.3390/healthcare10071202

**Published:** 2022-06-27

**Authors:** Jaewon Lee, Jaeho Yu, Jiheon Hong, Dongyeop Lee, Jinseop Kim, Seonggil Kim

**Affiliations:** Department of Physical Therapy, Sunmoon University, Asan 31460, Korea; naresa@sunmoon.ac.kr (J.Y.); hgh1020@sunmoon.ac.kr (J.H.); leedy@sunmoon.ac.kr (D.L.); skylove3373@sunmoon.ac.kr (J.K.); sgkim4129@sunmoon.ac.kr (S.K.)

**Keywords:** augmented reality, proprioceptive exercise, balance, positioning sensation, flexibility

## Abstract

This study investigates whether Augmented Reality (AR)-based interventions can be as effective as physical therapists (PT) regarding balance, positioning sensation, and flexibility. A sample of 39 regular people who voluntarily participated in this study were randomly distributed into two groups. Then AR was applied in the experimental group and PT was applied in the control group. Variables were measured by Tetrax(static balance), Y-balance test (dynamic balance), CSMI (proprioception), and sit and reach test (flexibility). All measurements were analyzed using paired *t*-test and independent *t*-test. The exercise program of this study improved the stability index (ST) of the static balance in both groups after the intervention, and there was a significant difference (*p* < 0.05) at normal eye close (NC) and Pillow with eye close (PC) positions. Moreover, regarding the case of dynamic balance, there were significant differences in AR and PT groups to reach in all directions (*p* < 0.05). In the case of positioning sensation, there was no significant difference in both groups (*p* > 0.05), and there was a significant difference in flexibility (*p* < 0.05). When comparing the two groups, there was no significant difference in all categories (*p* > 0.05). As a result, AR can be considered an effective form of therapy and can be selected according to individual conditions.

## 1. Introduction

Augmented reality is a computer technology that brings digital information into reality and makes it seem as if it were in its original environment [1]. This is a different concept from virtual reality (VR) because it exists and interacts in time and space like reality. Furthermore, advances in science and technology have made digital information, reality, and interaction between users smoothers. As a result, research is being conducted in various fields such as games and medical care, etc., and it is affecting society as a whole [2].

In particular, active research is underway in the medical field. The most frequently studied subjects in diagnosis, surgery, rehabilitation, education, and training, including pain, stroke, Parkinson’s disease, Alzheimer’s disease, and degenerative neuropathy. This medical condition reduces balance and increases the risk of falling due to instability. However, continuous research is needed to determine whether AR-based rehabilitation is as effective as a physical therapist (PT)’s rehabilitation [3].

Many studies are underway on whether AR-based rehabilitation can achieve the same effect as a PT. When comparing the improvement of balance ability by dividing 76 patients with Parkinson’s disease into two groups of exercise suggested by AR and proprioceptive exercise performed by a therapist, mobility, static, and dynamic balance (posture control) improved in both groups, and the average improvement in PT groups was greater. However, the actual difference between the two groups was minimal [4]. In addition, AR-based exercise in groups with significantly reduced balance, such as the older people, stroke, and Parkinson’s disease, showed as much improvement in gait and balance as the therapist [5,6,7].

Balance is the ability of the human body to maintain the position of COG within the BOS and process and execute input of visual, vestibular system, and somatic senses. And balance is essential to maintain stability from a static posture to a dynamic stance [8]. The body’s equilibrium to maintain balance consists of inputs from three systems: 70% propitious system, 20% vestibular system, and 10% visual system, while the somatosensory system processes proprioceptive information input from spin-cerebellar paths [9].

In many studies, proprioceptive exercise was carried out using tools such as BOSU and Swiss Ball to create the unstable ground, and a six-week proprioceptive program aimed at the older people greatly improved core stability and balance [10]. In addition, proprioceptive exercise in patients with chronic ankle instability enhanced static and dynamic balance [11]. Proprioceptive Neuromuscular Facility (PNF) stretching stimulates proprioceptors such as Golgi Tendon Organs (GTO) and Muscle Spindle located in muscles, tendons, and joints, and GTO activation inhibits muscle contraction. Neurophysiological mechanisms based on proprioceptive signals from these mechanical receptors can improve muscle length recovery and ROM, and as a result, it prevents biomechanical changes by relieving muscle tension. In the study comparing the effects of static, PNF, and mulligan stretching, the mulligan and hold-relax techniques of PNF stretching were effective [12]. The proprioceptive exercise program that includes relaxation exercises such as stretching, slow walking, etc. during warm-up, and cool down exercises, greatly improve balance, muscle strength, and flexibility in old people and sitting women studies [10,13,14].

As a such, AR-based and Swiss Ball exercises greatly improved the patient’s balance and significantly reduced the risk of falling due to instability. However, there was no study to confirm the improvement of flexibility by applying both Swiss Ball and AR-based proprioceptive exercise. Therefore, this study aims to find out the effects of AR on its effectiveness and balance, positional sense, and flexibility.

## 2. Materials and Methods

### 2.1. Participants

This study was controlled on 42 regular adults. The study subjects obtained the number of samples using the sample count calculation program ‘GPOWER, 3.1.9.7’, and the contents are as follows [15], (Table 1).

Participants were recruited through the local community website and university announcements. The participants listened to the purpose and method of the study and volunteered to fill out a written consent form before participating in the study. Participants in the study were selected as those who did not have a history of surgery or orthopedic surgery in the last six months. They did not have pain limiting current exercise performance. In addition, if any reasons could negatively affect the research process, they were excluded from the experiment: Neurological disease, cognitive damage, vestibular organ abnormalities, those who are taking drugs related to muscle strength and mental illness, and those who can no longer participate in the experiment due to COVID-19 during the experiment. Before the experiment, all participants measured their height and weight using the automatic BMI measuring instrument (BSM370, Inbody, Seoul, Korea, 2011). The general characteristics of participants are as follows (Table 2).

The study was set to Single-Blind, where participants did not know which group, they belonged to and were randomly divided into two groups before the experiment. The Institutional Review Board (IRB) of Sunmoon University approved this study (SM-202104-024-2). All collected data are kept safe by the Research Director.

### 2.2. Experiment Procedures

The flow of research is as follows (Figure 1). All participants were randomly distributed into two groups through an Excel function. The experimental group performed an augmented reality-based proprioceptive exercise program (ARPE), and the control group performed a proprioceptive exercise program conducted by a physical therapist (PTPE). All participants measured general characteristics such as age, height, and leg length before the experiment. In addition, variables of static balance, dynamic balance, positional sense, and flexibility were measured before the experiment and were measured once more after the experiment. 

### 2.3. Measurement Tools and Methods

All study participants measured their height and weight once before the experiment. Moreover, static, dynamic balance, positioning sensation, and flexibility are measured twice before and after the experiment using Tetrax, Y-Balance tests (YBT), CSMI, and sit and reach tests. The same researcher conducts all measurements and evaluations to reduce the error range.

#### 2.3.1. Measuring Stability of Body (Static Balance)

Static balance is measured using Tetrax (Tetra-ataxiometric posturography, Israel). Tetrax consists of computers and measuring instruments, measuring the pressure given to the measuring instrument’s four force plates (A, B, C, D), digitizing it, and transmitting it to the Tetrax computer’s software program. The computer analyzes the interaction and coupling phenomenon between each plate and analyzes various causes of the balance problem. First, the measurer causes the subject to go up barefoot following the foot’s shape depicted on the plate. Afterward, the measurer instructed the subject to stare in front of them and asked them to stand comfortably with arms on their sides without speaking or moving during the measurement. For the measurement, normal eye open (NO), normal eye close (NC), Pillow with eye open (PO), and Pillow with eye close (PC) were measured four times in order, and the unstable ground was prepared using a filler between the foot and the footplate (Figure 2). All measurements were measured twice before and after intervention for 32 s for each position. If the subject showed any movement that hindered the result during the measurement, the measurement was discarded and re-measured. The stability index (ST) and weight distribution index (WDI) were evaluated. ST is an index of overall stability, indicating how shaky the object is during measurement. WDI is an index showing weight distribution as a percentage, and the larger the value, the more the change in weight distribution changes than the natural value of 25%. These two values mean that the higher the balance, the lower it is [16,17].

#### 2.3.2. Measuring Stability of Body (Dynamic Balance)

Dynamic balance measures the functional reach distance of the lower extremity with a Y-balance test. During the measurement, the subject took off their shoes to minimize the factors affecting balance. Then, the subject placed their hand on the pelvis, stood on one foot on the central axis, and measured the value by pushing out the measuring instrument located in the anterior, posteromedial, and posterolateral with the opposite foot. Subjects practiced once in each direction and measured three times in each direction. Finally, the maximum reach was measured by reading the scale where the tip of the big toe and the edge of the measuring instrument meet (Figure 3). If the starting posture was not maintained during a performance or measured differently from the actual distance reached by pushing the measuring instrument hard, the phase was discarded and measured again. Calculate the maximum reach using the following formula {(sum of 1, 2 and 3 times reach)/(3 × leg length) × 100} [18].

##### Measuring Leg Length

Leg length is measured in a position where the subject lies directly on a flat table. The measurer manually pulls the subject’s leg to align the pelvis and measures the subject’s dominant leg in cm from ASIS to the most protruding part of the medial malleolus using a tape measure [19].

#### 2.3.3. Positioning Sensation Test

Positioning sensation is measured using isokinetic measuring equipment (CSMI, Humac norm, Computer Sports Medicine Inc, MA, USA, 2010). First, the measurer provides education on the knee joint flexion and extension movement to the subject and then randomly sets the subject’s leg position. After that, the subject is instructed to remember the initial location, and the subject is required to create the initial target angle with their eyes closed (Figure 4). Measurements are made three times to reduce errors, and the average value is calculated [20].

#### 2.3.4. Measuring Flexibility of Body

Flexibility is measured by sit and reach tests. A scale is marked cm at the top of the box inspection tool, and a movable measuring instrument is marked. The start of the inspection tool is the standard (0 cm). The subject sits with their legs straight and puts their bare feet on the side of the inspection tool. The measurer teaches the measurement method, and the subject gathers their hands after one practice, slowly bends forward, pushes the measuring instrument, maintains it for a short period, and returns to its original position. The maximum reach was measured by reading the scale of the tip of the subject’s 3rd finger and the edge of the measuring instrument. At this time, if the subject’s knee is bent, the measuring instrument is pushed by the recoil of the trunk, or the end posture is not maintained, the phase is discarded and measured again. Measurements are made three times to reduce errors, and the average value is calculated [21].

### 2.4. Intervention Method

ARPE follows augmented reality (AR), and PTPE follows the exercise program conducted by the physical therapist. Before the experiment, the researcher explained to all participants about light education and research procedures for exercise programs and suggested that they wear comfortable clothes for accurate and safe experiments. The researcher checked the laboratory environment at all times and maintained it under the same conditions as safe.

#### Proprioceptive Exercise Program

The exercise program consists of the following (Figure 5), (Table 3). ARPE follows AR’s verbal instructions and visual feedback, and PTPE follows the therapist’s instructions and feedback. The proprioceptive exercise program was held twice a week for four weeks and was applied for 35 min. For the first two weeks, each exercise was applied 10 times per set, 2 sets, and after that, 3 sets of 12 times were applied by increasing the intensity. The Swiss ball was provided in a size of 55 to 65 cm according to the height of the study participant and proceeded barefoot to minimize the variable.

### 2.5. Data Analysis

All statistical analyses used ‘IBM SPSS 26.0 Statistical Software’. For each variable, general characteristics were evaluated using descriptive statistics, and the mean (M) and standard deviation (SD) were calculated. Paired *t*-test was used for pre and post variables within each group, and an independent *t*-test was used to compare the results between the two groups. The statistical significance level was set to α = 0.05.

## 3. Results

### 3.1. Static Balance

In the pre-measurement, there was no significant difference in static balance in all variables (*p* > 0.05) (Figure 5), (Table 4). After the intervention, the pre- and post-mortem results of each group showed no significant difference in variables other than NC and PC of ST in both groups (*p* > 0.05) (Table 5). However, there was a significant difference between PC and NC of ST (*p* < 0.05). And there was no significant difference in the post-measurement between the two groups (*p* > 0.05) (Figure 5), (Table 4).

### 3.2. Dynamic Balance

As a result of YBT pre-measurement, there was no significant difference between groups in ANT, PM, and PL directions (*p* > 0.05) (Figure 6), (Table 6). After the intervention was applied, the measurement results within each group showed significant differences in the direction of ANT, PM, and PL in both groups (*p* < 0.05) (Table 7). However, there was no significant difference in post-measurement results between the two groups (*p* > 0.05) (Figure 6), (Table 6).

### 3.3. Positioning Sensation & Flexibility

As a result of the pre-measurement of positioning sensation, there was no significant difference between groups (*p* > 0.05), and even after the intervention, no significant difference could be identified within the group. Likewise, there was no significant difference between groups in the post-measurement results (*p* > 0.05) (Figure 7), (Table 8).

In the case of flexibility, there was no significant difference between groups as a result of pre-measurement (*p* > 0.05). After the intervention was applied, a significant difference was confirmed between the two groups (*p* < 0.05), there was no significant difference between the two groups as a result of post-measurement(*p* > 0.05) (Figure 7), (Table 9).

## 4. Discussion

People who have problems with static balance are more dependent on proprioception than normal people. So, proprioceptive exercise is important in the rehabilitation process and needs to be designed appropriately [22]. We have conducted balance training using unstable training tools such as balance boards, and these have been effective in stimulating proprioception [10,11]. Research based on augmented reality (AR), as well as existing educational tools, is being conducted. A four-week of AR exercise for older people showed improvements in lower limb function and balance, lowering the risk of falling, Im et al. reported [23]. This suggests that AR alone can improve balance in general exercise therapy. This study confirmed that there was no significant difference between the two groups. In the case of regular people, it is difficult to determine the difference in effects without external stimulus, but due to changes in the environment, such as blocking visibility and providing unstable support, the dependence on unique perception is further increased [24]. Hebner (2021) confirmed that the contribution of proprioception varies with the presence or absence of time when adjusting standing balance under various conditions [25]. Previous studies prove that it can be affected by the presence or absence of vision and confirm that NC and PC of ST have improved in this study.

In addition, the exercise program of this study was also involved in improving dynamic balance and is related to YBT performance ability and muscle strength. Jaber et al. (2018) reported that the reduction of hip muscles affects the ability to maintain balance, negatively affecting functional movement [26]. Nelson et al. (2021) demonstrated that neuromuscular regulation and intensity of the quadriceps group are important predictors of ANT arrival performance, and asymmetry of ANT arrival distance is associated with an increased risk of various limb injuries [27]. In addition, according to Kang et al. (2015), the balanced movement obtained by trunk stabilization and extension is correlated with ANT’s maximum reach, and trunk flexion also has a positive effect on the maximum distance that can be reached backward [28]. Wilson et al. (2018) announced that PM and PL reach are related to backward muscle strength, including hip extensor moment [29]. In conclusion, the trunk, hips, and knees movement are all-important to YBT reach, and knee extensors in ANT and PM and hip extensors in PL and PM are the most notable factors. The exercise program in this study consisted of the exercise of these muscle groups, showed dynamic balance improvement in all directions, and was consistent with those reported in previous studies. In AR-based research, the results of YBT are not significantly different. The eight-week Otago exercise for middle-aged adults improved dynamic balance by improving hip muscles and pelvic stability. This indicates that the performance of AR-based strengthening and balance exercises is not different from the results of previous exercises performed by therapists [30]. In this study, no significant difference was found between groups.

Muscle strength required to maintain balance is less when stationary than in functional movement. QipengSong et al. (2021) indicated that muscle strength was correlated with dynamic balance but not static balance [31]. In addition, hip muscles correct large errors, while ankle muscles correct minor errors. Since the COP continues to change due to repulsive ground force, it is suggested that the ankle strategy, which is considered to help reduce the angle of swing of ankle torque, is a good strategy during static balance [32]. However, this study does not include exercise targeting only the ankle, so it is expected that there was no significant difference in WDI.

Proprioception is a complex entity that includes a positioning sensation, a velocity sensation, and a sense of force to perform this movement in space [33]. Positioning sensation is measured based on accuracy in reproducing the target angle [34]. In this study, positioning sensation was improved than before, but it was not easy to identify statistically significant differences. For this reason, participants were recruited as young, healthy adults with no problems with proprioception. In addition, the stretching techniques applied to the two groups may not have been sufficient to apply time or tension enough to make a difference in positional sensations to all machine receptors around the knee. Aslan et al. (2018) immediately after applying HR-PNF, re-measurements were made over time and finally compared with the values before and after, confirming that the effect decreased over time. Additionally, HR-PNF application is also related to the relationship between the characteristics of YBT and the direction of arrival. It is explained as a decrease in reciprocal inhibition of gluteus maximus, and the difference between directions was maintained over time after stretching. In particular, it showed a significant improvement in PM and PL [35]. Mani et al. (2021) applied static stretching and PNF stretching to the hamstring immediately and long-term to compare positional sensations, and immediately they were similar to each other. In the long run, PNF further improved the positioning sensation [36]. This study was applied shorter than the previous study due to its four-week application. However, it is expected that it can be significantly improved when applied in the long run.

As mentioned earlier, stretching is effective in improving flexibility. In addition, stretching is essential to restore optimal muscle length. In particular, it has been reported that the tension of the hamstring causes biomechanical changes and negatively affects dynamic balance. Therefore, warming up, which includes stretching, results in beneficial results in exercise performance [37]. Salami et al. (2021) announced that static stretching of the hamstring could have a positive effect on improving balance ability as well as increasing flexibility [38]. Furthermore, Mani et al. (2021) compared static and PNF stretching in the short and long term and found that both groups greatly improved flexibility [36]. Based on the results of previous studies, it can be confirmed that the application of warming up, including stretching in this study, greatly improved flexibility and had a positive effect on balance. Miklaus et al. (2021) announced that combining high-level exercise, including visual feedback, improves cognitive function and effectively controls motion. Exercise in virtual reality (VR), including visual feedback, has created a favorable environment for rehabilitation of the lower extremity of chronic patients after stroke and has better results in straight leg raise growth, which correlates flexibility with ROM [39]. AR equipment can also have the same effect as VR because it recognizes the subject’s body, shows the subject’s location through the screen, and analyzes and provides the patient’s posture and movement at the same time as visual feedback. The effect of virtual and AR-based strengthening exercises on physical performance is explained by the social cognitive theory mechanism by meta-analysis. In other words, when subjects exercise in AR, they can observe and learn their action without rewards and punishment, and when their illuminated self-image and virtual models are similar to themselves, they are more likely to learn the results of their actions [40]. Based on the experimental results of this study and the results of previous studies, AR has similar effects to PT.

This study has several limitations. This study has several limitations. It is challenging to generalize compared to various age groups because it targets healthy adults in their 20 s. In addition, there were restrictions on controlling the daily life of the subjects. However, as the results of this study and previous studies support, exercise (intervention) using AR can have a similar effect as a physical therapist. AR is easily accessible in areas such as islands and mountains that are medically alienated, and it is a new motivation for exercise. In addition, AR can overcome home limitations and improve exercise accuracy by providing immediate feedback without a therapist.

## 5. Conclusions

The purpose of this study is to confirm whether augmented reality (AR)-based interventions can be as effective as therapists, and the results are as follows. First, the exercise of this study can improve flexibility and significantly improve balance ability to prevent injuries caused by falls. Second, both PT and AR interventions effectively improve balance and flexibility. And AR does not differ in balance and flexibility compared to the treatment of therapists, allowing them to choose their preferred treatment in individual environments. As a result, it can be concluded that AR-based interventions can be as effective as therapists, and that some interventions can be helped by replacing therapists.

## Figures and Tables

**Figure 1 healthcare-10-01202-f001:**
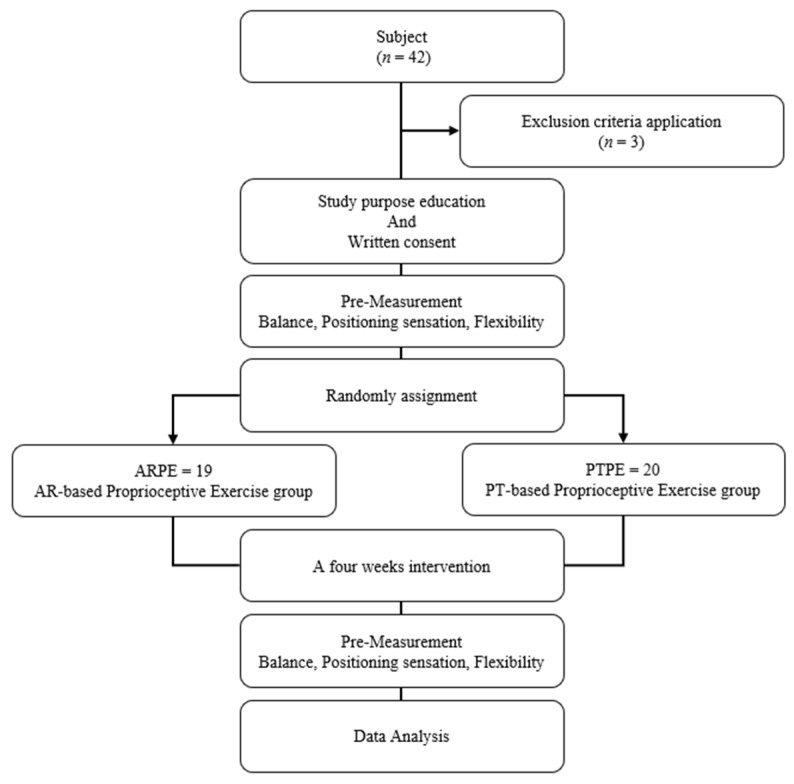
Experiment Procedures.

**Figure 2 healthcare-10-01202-f002:**
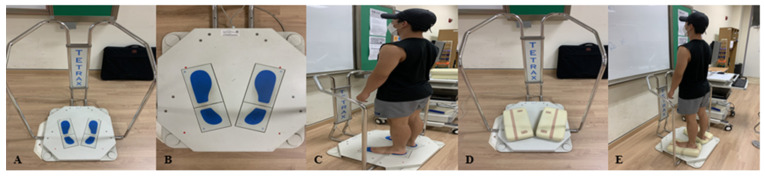
Static balance measurement: (**A**) Tetrax measuring instrument; (**B**) force plate; (**C**) measure on a flat ground; (**D**) unstable ground with a pillow; (**E**) measure on an unstable ground.

**Figure 3 healthcare-10-01202-f003:**
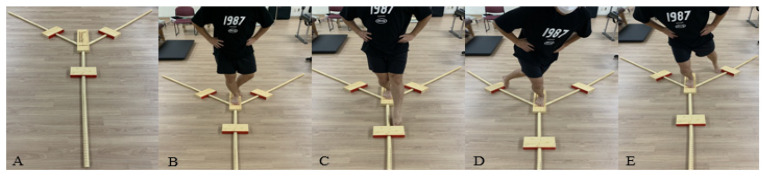
Y-balance test. (**A**) Y-balance test equipment; (**B**) starting posture; (**C**) anterior; (**D**) posteromedial; (**E**) posterolateral.

**Figure 4 healthcare-10-01202-f004:**
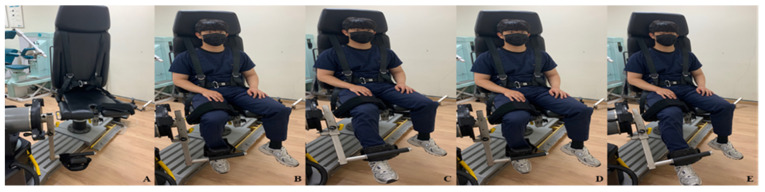
CSMI. (**A**) CSMI; (**B**) starting position; (**C**) set leg to target angle; (**D**) back to (**B**) and close eyes; (**E**) make a target angle.

**Figure 5 healthcare-10-01202-f005:**
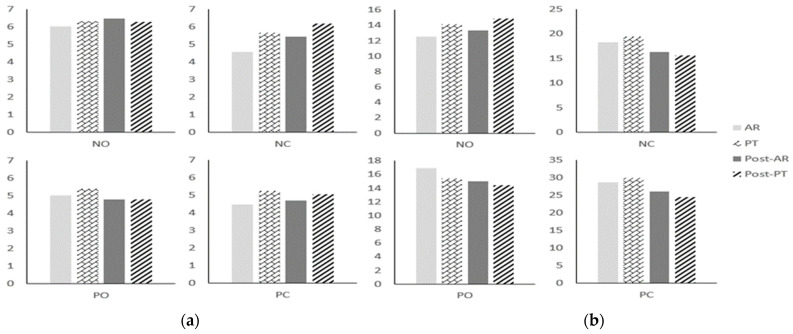
Comparison of static balance between groups after intervention: (**a**) WDI; (**b**) ST.

**Figure 6 healthcare-10-01202-f006:**
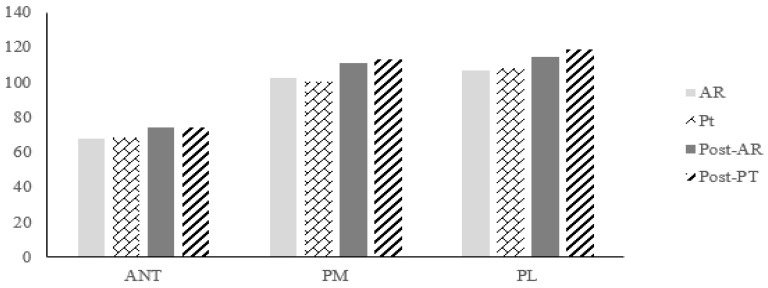
Comparison of dynamic balance between groups after intervention. AR: Pre-measurement of the experimental group; PT: Pre-measurement of the control group; Post-AR: Post-measurement of the experimental group; Post-PT: Post-measurement of the control group.

**Figure 7 healthcare-10-01202-f007:**
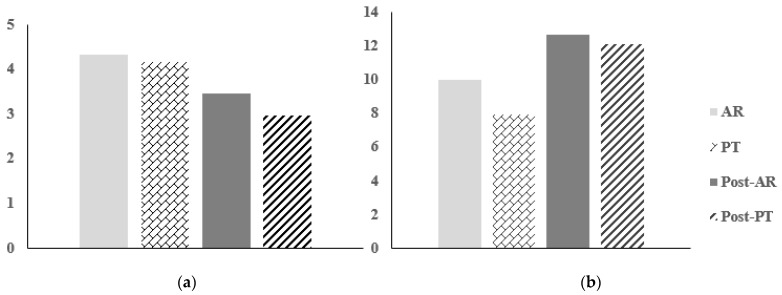
Comparison of positioning sensation and flexibility between groups after intervention application: (**a**) Positing sensation; (**b**) Flexibility.

**Table 1 healthcare-10-01202-t001:** Detailed data values for calculating the number of samples.

Test Family	T Test
Type of Power Analysis	A priori: compute required sample size—given α, power, and effect size
Effect Size d	0.8
α err prob	0.05
Power (1-B err prob)	0.8
Allocation ratio N2/N1	1
Total Sample Size	42

**Table 2 healthcare-10-01202-t002:** General characteristics of participants.

	ARPE (*n* = 19)	PTPE (*n* = 20)
Age (years)	22.67 ± 2.90	21.76 ± 1.41
Height (cm)	166.43 ± 22.12	166.91 ± 19.24
Weight (Kg)	78.33 ± 30.31	64.26 ± 14.29

Values indicate mean ± standard deviation, ARPE: augmented reality proprioceptive exercise, PTPE: physical therapy proprioceptive exercise.

**Table 3 healthcare-10-01202-t003:** Proprioceptive exercise program.

Types of Exercise	Explanation
Warming–Up (Stretching)	Stretching is performed on Hamstrings that have the biggest impact on “sit and reach test” and for calf and hip muscles. ARPE proceeds with static stretching, and a therapist performs PTPE with a Hold-relax technique of PNF stretching. Stretching progresses for 15 s in both groups.
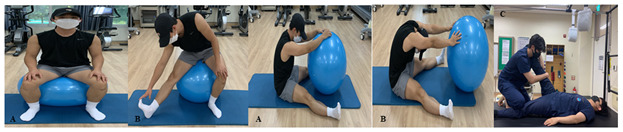
Swiss Ball Straight arm crunch	First, the subjects are to put up their knees and lie down properly. Then raise the Swiss ball above themselves and extend their legs forward to the ball. Afterwards, they slowly lift their upper body, pushes the ball up, and returns to the starting position.
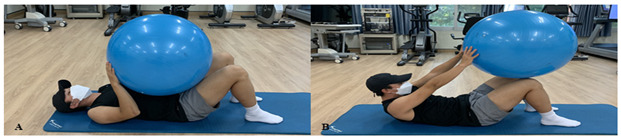
Swiss Ball alternate arm and leg flexion	The subject sits on the ball and bends their knees 90 degrees. Lift the left leg and right arm at the same time and return to the starting position. After that, do the opposite leg and arm.
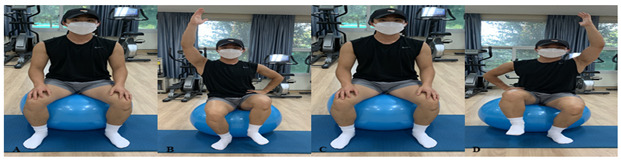
Swiss Ball shoulder bridge	The subject lies in the correct position, bends his knees 90 degrees, and puts his soles on the ball. Slowly lift their hips so that they are in line with their shoulders while preventing the ball from being pushed back to the starting position.
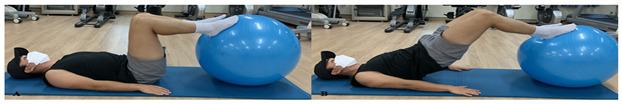
Swiss Ball back extension	The subject lies on their stomach on the ball, straightens their knees, and lifts their upper body and arm upward (backwards). After that, they come back to the starting position.
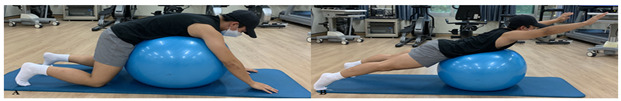
Swiss Ball hamstring curl	The subject lies comfortably on their back, straightens their legs, and puts their heels on the ball. Using Hamstring, slowly bend their knees and pull the ball so that the soles of their feet touch. After that, they come back to the starting position.
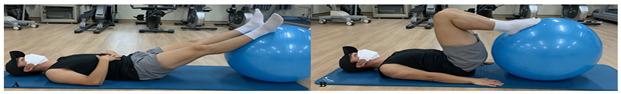
Swiss Ball leg raise	The subject lies upright, puts their palms on the floor, bends their knees, and puts the ball between their ankles. Then, put strength on their abdomen and stretch their knees to lift the ball up. After that, they come back to the starting position.
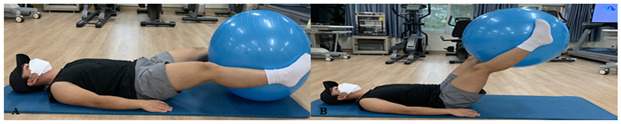

A: starting position, B: end position, C, D: ongoing posture.

**Table 4 healthcare-10-01202-t004:** Comparison of static balance between groups after intervention.

		Pre			Post		
		ARPE	PTPE	*t*	ARPE	PTPE	*t*
**WDI**	**NO**	6.02 ± 2.64	6.32 ± 3.74	−0.285	6.46 ± 2.87	6.27 ± 2.90	0.199
	**NC**	4.55 ± 2.17	5.67 ± 3.25	−1.261	5.45 ± 2.96	6.16 ± 2.26	−0.839
	**PO**	5.02 ± 1.99	5.40 ± 3.37	−0.433	4.77 ± 2.86	4.77 ± 2.74	−0.004
	**PC**	4.47 ± 2.56	5.23 ± 3.31	−0.795	4.70 ± 2.16	5.06 ± 2.59	−0.480
**ST**	**NO**	12.53 ± 2.96	14.18 ± 4.72	−1.300	13.33 ± 3.54	14.85 ± 5.42	−1.032
	**NC**	18.21 ± 5.62	19.41 ± 8.22	−0.528	16.34 ± 5.02	15.61 ± 4.60	0.471
	**PO**	16.88 ± 4.27	15.39 ± 6.09	0.883	14.99 ± 3.07	14.45 ± 4.44	0.436
	**PC**	28.67 ± 6.67	29.90 ± 11.51	−0.407	26.03 ± 5.22	24.42 ± 7.23	0.790

Mean ± standard deviation, ARPE: augmented reality proprioceptive exercise, PTPE: physical therapy proprioceptive exercise, ST: stability index, NO: normal eye open position, NC: normal eye close position, PO: pillow with eye open position, PC: pillow with eye close position, WDI: weight distribution index.

**Table 5 healthcare-10-01202-t005:** Comparison of static balance within a group after intervention.

			Pre	Post	*t*
**ARPE**	**WDI**	**NO**	6.02 ± 2.64	6.46 ± 2.87	−0.636
		**NC**	4.55 ± 2.17	5.45 ± 2.96	−1.778
		**PO**	5.02 ± 1.99	4.77 ± 2.86	0.610
		**PC**	4.47 ± 2.56	4.70 ± 2.16	−0.457
	**ST**	**NO**	12.53 ± 2.96	13.33 ± 3.54	−1.141
		**NC**	18.21 ± 5.62	16.34 ± 5.02	2.593 *
		**PO**	16.88 ± 4.27	14.99 ± 3.07	2.026
		**PC**	28.67 ± 6.67	26.03 ± 5.22	2.252 *
**PTPE**	**WDI**	**NO**	6.32 ± 3.74	6.27 ± 2.90	0.058
		**NC**	5.67 ± 3.25	6.16 ± 2.26	−0.803
		**PO**	5.40 ± 3.37	4.77 ± 2.74	0.998
		**PC**	5.23 ± 3.31	5.06 ± 2.59	0.333
	**ST**	**NO**	14.18 ± 4.72	14.85 ± 5.42	−0.825
		**NC**	19.41 ± 8.22	15.61 ± 4.60	3.249 **
		**PO**	15.39 ± 6.09	14.45 ± 4.44	1.384
		**PC**	29.90 ± 11.51	24.42 ± 7.23	3.526 **

* *p* < 0.05, ** *p* < 0.01, mean ± standard deviation, ARPE: augmented reality proprioceptive exercise, PTPE: physical therapy proprioceptive exercise, ST: stability index, NO: normal eye open position, NC: normal eye close position, PO: pillow with eye open position, PC: pillow with eye close position, WDI: weight distribution index.

**Table 6 healthcare-10-01202-t006:** Comparison of dynamic balance within a group after intervention.

		Pre	Post	*t*
**ARPE**	**ANT**	67.50 ± 5.81	74.35 ± 7.00	−5.320 **
	**PM**	102.20 ± 10.78	111.23 ± 9.04	−5.183 **
	**PL**	106.36 ± 12.28	114.45 ± 8.09	−3.343 **
**PTPE**	**ANT**	68.23 ± 7.33	73.88 ± 6.24	−3.554 **
	**PM**	100.14 ± 16.96	113.25 ± 11.37	−3.793 **
	**PL**	107.87 ± 14.50	119.01 ± 12.50	−4.254 **

** *p* < 0.01, mean ± standard deviation, ARPE: augmented reality proprioceptive exercise, PTPE: physical therapy proprioceptive exercise, ANT: Anterior, PM: Posteromedial, PL: Posterolateral.

**Table 7 healthcare-10-01202-t007:** Comparison of dynamic balance between groups after intervention.

	Pre			Post		
	ARPE	PTPE	*t*	ARPE	PTPE	*t*
**ANT**	67.50 ± 5.81	68.23 ± 7.33	−0.345	74.35 ± 7.00	73.88 ± 6.24	0.218
**PM**	102.20 ± 10.78	100.14 ± 16.96	0.454	111.23 ± 9.04	113.25 ± 11.37	−0.612
**PL**	106.36 ± 12.28	107.87 ± 14.50	−0.351	114.45 ± 8.09	119.01 ± 12.50	−1.359

Mean ± standard deviation, ARPE: augmented reality proprioceptive exercise, PTPE: physical therapy proprioceptive exercise, ANT: Anterior, PM: Posteromedial, PL: Posterolateral.

**Table 8 healthcare-10-01202-t008:** Comparison of positioning sensation within and between groups before and after intervention application.

	ARPE	PTPE	*t*
Pre	4.33 ± 3.27	4.15 ± 2.44	0.199
Post	3.46 ± 2.19	2.95 ± 1.82	0.786
*t*	0.978	1.836	

Mean ± standard deviation, ARPE: augmented reality proprioceptive exercise, PTPE: physical therapy proprioceptive exercise.

**Table 9 healthcare-10-01202-t009:** Comparison of flexibility within and between groups before and after intervention application.

	ARPE	PTPE	*t*
Pre	9.96 ± 9.35	7.90 ± 13.69	0.552
Post	12.63 ± 9.10	12.08 ± 10.90	0.170
*t*	−2.996 **	−3.911 **	

** *p* < 0.01, mean ± standard deviation, ARPE: augmented reality proprioceptive exercise, PTPE: physical therapy proprioceptive exercise

## Data Availability

The data used to support the findings of this study are available from the corresponding author upon request.

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
