# Peer review of "The Effect of Augmented Reality-Based Proprioceptive Training Program on Balance, Positioning Sensation and Flexibility in Healthy Young Adults: A Randomized Controlled Trial"

_healthcare, 2022, doi:10.3390/healthcare10071202_

Round 1

Reviewer 1 Report

Despite the important limitation of the very limited age range of the participants, both the methodological quality and the results obtained with the study represent an manuscript of sufficient interest for publication in this journal

Author Response

Thank you for your comments.

Reviewer 2 Report

In this particular manuscript, the authors present an interesting Augmented Reality-Based Proprioceptive Training Program and its effects on balance, positioning sensation and flexibility.

However, I have some suggestions:

Point 1. Title: it is not clear. I suggest to add that the training program is addressed to healthy young adults

Point 2. Introduction

1. Please correct Mobility (line 44) to mobility

2. Explain the abbreviation GTO (line 61)

3. How come Neurophysiological mechanisms based on proprioceptive signals from these mechanical receptors can improve muscle length recovery and ROM, and as a result, posture and stability are greatly affected” (line 62-64). If muscle length and ROM improve, why posture and stability are affected?

4.The proprioceptive exercise program, which includes stretching in warming-up  (line 66). The program incudes only stretching warm-up?

Point 3. Materials and Methods

1. It is mandatory to register with a clinical trial registry, if the study involves human subjects with prospective recruitment... Is this a prospective cohort study, or a clinical trial?

2. Please correct stability index (ST) (line 134) to Stability index

3. 2.4 Intervention method

Include the details of the Augmented Reality-Based Proprioceptive Training Program. It is not clear how the subjects perform this intervention.

Proprioceptive exercise program (line 184). It is not mentioned to which muscles, the hold-relax technique of PNF stretching, is addressed? trunk, hips, and knees muscles?

Point 4. Discussion

The authors present several studies related to the topic proposed in this article but do not highlight their results.

Author Response

Your advice gave more specific information to the reader, and it was a great help to me too. Thank you for your comments. Please refer to the red letters.

Point 1Title: Modification completed.

Point 2Introduction

  1. Modification completed.
  2. Modification completed.
  3. It seems that the meaning was not clearly conveyed during the translation. I modified it.
  4. It seems that the meaning was not clearly conveyed during the translation. I modified it.

Point 3. Materials and Methods

  1. a clinical trial
  2. Modification completed.
  3. Modification completed. This is simply a description of the composition of the exercise program. The exercise procedure is shown in line 102.
  4. Modification completed.

Point 4. Discussion

I think 'discussion' should describe why these results came out through previous papers on data interpretation according to the results of this study. I think the results of this study were fully emphasized in the 'results' and 'conclusions'.

Reviewer 3 Report

Dear authors, following are my recommendations for revision.

Abstract

Please write out Augmented Reality the first time and put AR as abbreviation in parenthesis.

Please put PT as abbreviation in parenthesis after the word physical therapists in the first sentence for further understanding.

Please include intervention group and control group as words in the text. Please iclude the data collection instruments.

What do the abbreviations ST, NC and PC mean in the abstract? Please write them out.

Please delete the limitations in the abstract.

You can shorten the sentences with the p values. Simply state significant and non-significant and add the p values in parentheses. Please do not embellish the sentences in order to have more space for writing out abbreviations.

You can delete the analysis statement in the abstract, unless it is a specific requirement of the journal.

Introduction:

Please avoid the word elderly in the introduction and in the whole text. It is discriminating. Please use e.g. older people.

Methods:

Please start first with the design of the study and the checklist you used for reporting (incl. citation of the checklist).

Participants, Line 77: Please correct the first sentence: " ....S University in A city".

Please include your recruitment strategy.

Experiment procedures: Please include the words intervention and control group.

Intervention method:

Please describe your used AR device for the intervention group and include a picture of it. Maybe it would also be interesting to see a picture with AR device in action.

Author Response

Your advice gave more specific information to the reader, and it was a great help to me too. Thank you for your comments. Please refer to the red letters.

Reviewer 4 Report

The authors of the article describe a study conducted on a sample of patients.

The study aims to demonstrate that applications that take advantage of augmented reality can help a patient's treatment phase.

The article is well structured and adequately describes the analysis that the authors have carried out.

The article needs English revision as some parts have errors or sentences that are not easily understood.

For example, the sentence on lines 77-80 should be reworded

In line 341, one sentence is repeated twice.

The sentence in line 356 is poorly worded.

The article needs a table that summarizes the acronyms used in order to make the text more accessible for the reader

Author Response

Your advice gave more specific information to the reader, and it was a great help to me too. Thank you for your comments. Please refer to the red letters.

I'll try to fix the English problem a little more.

Round 2

Reviewer 2 Report

Point 3. Materials and Methods

It is mandatory to register in a clinical trial registry, if the study involves human subjects with prospective recruitment.
Intervention method : Include the details of the Augmented Reality-Based Proprioceptive Training Program. It is not clear how the subjects perform this intervention.

Author Response

Please refer to the red letters.  1. It was confirmed that it was approved by IRB and is currently in the process of research registration. It will be completed as soon as soon as possible. 2. The contents of the experimental procedure were transferred to the intervention method. The detailed exercise items' description and FITT were accurately specified. In addition, exercise intervention was said throughout the paper that only the subject who conducts exercise was different, and I think it was explained enough to be understood and performed by a third party.   Thanks for your comments :)

Reviewer 3 Report

Dear Authors, I have no further recommendations.

Author Response

Thank you for your comments.
